# ThinkGeo 🌐 : Evaluating Tool-Augmented Agents for Remote Sensing Tasks

## Abstract

Recent progress in large language models (LLMs) has enabled tool-augmented agents capable of solving complex real-world tasks through step-by-step reasoning. However, existing evaluations often focus on general-purpose or multimodal scenarios, leaving a gap in domain-specific benchmarks that assess tool-use capabilities in complex remote sensing use cases. We present **ThinkGeo**, an agentic benchmark designed to evaluate LLM-driven agents on remote sensing tasks via structured tool use and multi-step planning. Inspired by tool-interaction paradigms, ThinkGeo includes human-curated queries spanning a wide range of real-world applications such as urban planning, disaster assessment and change analysis, environmental monitoring, transportation analysis, aviation monitoring, recreational infrastructure, and industrial site analysis. Queries are grounded in satellite or aerial imagery, including both optical RGB and SAR data, and require agents to reason through a diverse toolset. We implement a ReAct-style interaction loop and evaluate both open and closed-source LLMs (e.g., GPT-4o, Qwen2.5) on 486 structured agentic tasks with 1,773 expert-verified reasoning steps. The benchmark reports both step-wise execution metrics and final answer correctness. Our analysis reveals notable disparities in tool accuracy and planning consistency across models. ThinkGeo provides the first extensive testbed for evaluating how tool-enabled LLMs handle spatial reasoning in remote sensing.

## 1 Introduction

Recent advances in LLMs have enabled the emergence of tool-augmented agents, systems that can break down complex tasks into step-by-step plans, invoke external tools (e.g., vision modules, calculators, and code interpreters), and reason across intermediate states Yao et al. (2023); Shen et al. (2023). This paradigm, popularized via ReAct-style frameworks Yao et al. (2023), has shown promise in general-purpose settings through benchmarks like ToolBench Qin et al. (2023), GAIA Mialon et al. (2023), and GTA Wang et al. (2024), which evaluate agents on procedural correctness, tool use, and final task outcomes. However, these benchmarks largely focus on synthetic, open-domain, or web-grounded scenarios, leaving the question of agentic capability in precision-critical, spatially grounded domains, like remote sensing, largely unexplored.

Remote sensing (RS) is critical to a wide range of applications, including environmental monitoring, urban infrastructure and transportation analysis, disaster response, and land-use mapping, with an ever-growing stream of high-resolution imagery from earth observation (EO) satellites and drones Kao et al. (2025). Despite advances in visual models for detection, segmentation, and change analysis, current processing pipelines remain brittle and manually engineered across tasks. Integrating these capabilities into LLM-driven agents demands reasoning over geodetic metadata, spatial resolutions, temporal dynamics, and unit-aware calculations. Existing agentic benchmarks (e.g., GTA Wang et al. (2024), GAIA Mialon et al. (2023)) do not address these demands; they are built around general-purpose or web-grounded images, lacking the spatial fidelity and grounding required for geospatial workflows. Consequently, there is a pressing need for a benchmark that evaluates tool-augmented agents in remote sensing contexts, for reasoning over real EO imagery, coordination of general-purpose visual tools, and handling spatially grounded multi-step tasks.

In this work, we introduce **ThinkGeo**, the first agentic benchmark specifically designed to evaluate tool-augmented LLM agents on realistic remote sensing tasks. As shown in Table 1, unlike exist-

Table 1: Comparison of agentic benchmarks across key dimensions. ThinkGeo is the only benchmark designed specifically for remote sensing (RS), incorporating real EO imagery alongside ReAct-style annotation chains and deployed tools. It uniquely supports spatial reasoning and remote sensing-specific tasks through geospatial grounded inputs and execution-level evaluation for studied models.

| Benchmark | Real queries | Deployed tools | MM inputs | Annotation chains | Execution eval. | RS images |
|---|---|---|---|---|---|---|
| API-Bench Patil et al. (2023) | ✗ | ✗ | ✗ | ✗ | ✗ | ✗ |
| ToolBench Qin et al. (2023) | ✗ | ✓ | ✗ | ✗ | ✗ | ✗ |
| GAIA Mialon et al. (2023) | ✓ | ✗ | ✓ | ✗ | ✓ | ✗ |
| APIBank Li et al. (2023) | ✗ | ✓ | ✗ | ✓ | ✗ | ✗ |
| m&m's Ma et al. (2024) | ✗ | ✓ | ✓ | ✓ | ✓ | ✗ |
| GTA Wang et al. (2024) | ✓ | ✓ | ✓ | ✓ | ✓ | ✗ |
| **ThinkGeo (Ours)** | ✓ | ✓ | ✓ | ✓ | ✓ | ✓ |

ing agentic benchmarks built on general or web-grounded images, ThinkGeo focuses on spatially grounded reasoning, requiring agents to plan and execute multi-step workflows using satellite and aerial imagery. Each query is coupled with an executable tool environment and annotated with structured evaluation signals, enabling rigorous assessment of perception, planning, and geospatial reasoning under tool-based execution constraints. Our main contributions are as follows:

- **Task Suite & Dataset:** A curated set of 486 agentic tasks with 1,773 expert-verified reasoning steps over medium to high-resolution optical RGB (with 436 tasks) and SAR (with 50 tasks) images, spanning urban, environmental, transportation, aviation, industrial, change detection, and disaster-related scenarios. Examples are shown in Figure 1.

- **Executable Tool Set:** An extended suite of 14 tools designed to simulate real-world RS workflows. This includes perception modules (e.g., `ObjectDetection`, `SegmentObjectPixels`, `ChangeDetection`), logic and numeric tools (e.g., `Calculator`, `Solver`, `Plot`), and visualization aids (e.g., `DrawBox`, `AddText`).

- **Evaluation Protocol:** We propose two evaluation modes, step-by-step and end-to-end, paired with fine-grained metrics to assess instruction adherence, tool use correctness, argument formatting, multi-step reasoning, and final answer accuracy.

- **Benchmarking Study:** A comparative evaluation of state-of-the-art LLM agents, including GPT-4o, Claude-3, Qwen-2.5, and LLaMA-3, revealing persistent gaps in multimodal tool reasoning and execution trace alignment, even among top-performing models.

By grounding agentic evaluation in real EO imagery and requiring interpretable, tool-based interaction tracking, ThinkGeo provides a new foundation for benchmarking and ultimately providing insights to improving spatially-aware, tool-augmented LLM agents for geospatial analysis.

## 2 RELATED WORK

**Tool-augmented LLM Agents and Benchmarks:** Integrating large language models (LLMs) with executable tools has recently become a central focus in agent research. Early work presented tool use as an alternating planning and execution. ReAct, for instance, interleaves "thought" tokens with structured tool calls, enabling a single LLM to both reason and act Yao et al. (2023). Subsequent systems generalized this idea to larger tool repertoires. HuggingGPT employs a GPT controller to select from hundreds of vision, speech, and language models exposed as functions Shen et al. (2023), while Visual ChatGPT and MM-ReAct demonstrate analogous pipelines for multimodal perception tasks Wu et al. (2023); Yang et al. (2024). To measure tool-use proficiency, several benchmarks have been proposed. ToolBench, APIBench, and API-Bench evaluate single-step API invocation within synthetic prompts Qin et al. (2023); Patil et al. (2023); Li et al. (2023); m&m's extends this to multi-step multimodal settings Ma et al. (2024). More recently, GAIA Mialon et al. (2023) and GTA Wang et al. (2024) introduced human-written, step-implicit tasks paired with executable tool

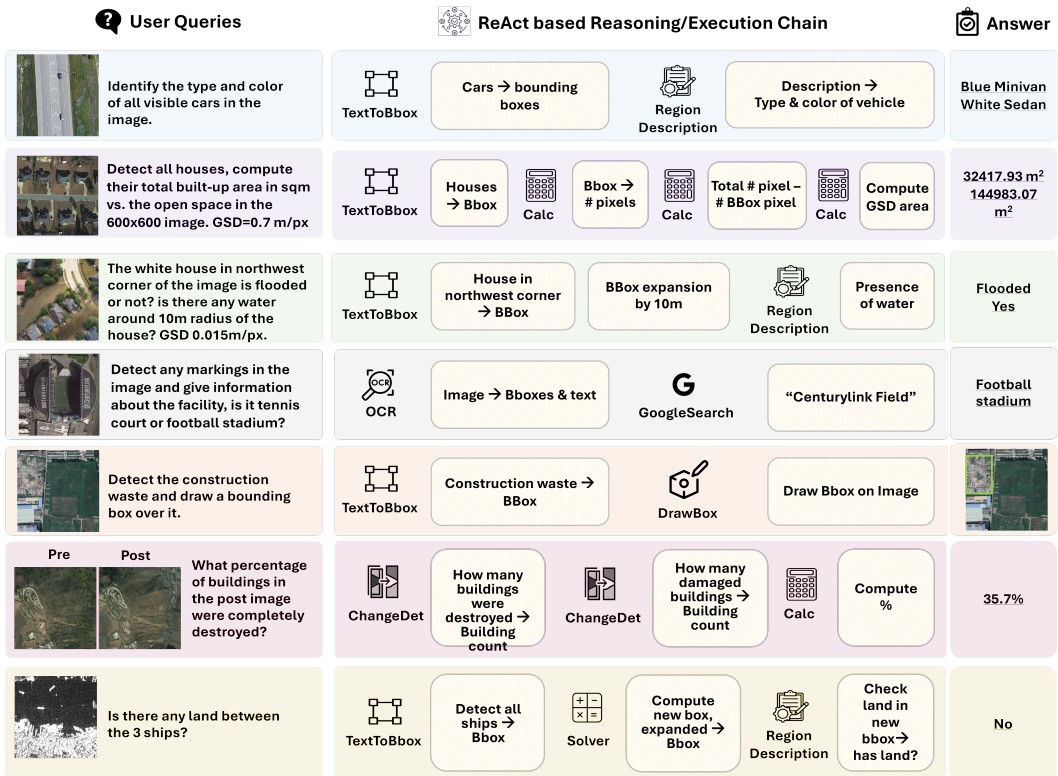

Figure 1: Representative samples from the ThinkGeo benchmark. Each row illustrates a user query grounded in real RS imagery, followed by a ReAct-based execution chain comprising tool calls and reasoning steps, and concludes with the resulting answer. The examples span diverse use cases, including transportation analysis, urban planning, disaster assessment and change analysis, recreational infrastructure, and environmental monitoring, highlighting multi-tool reasoning and spatial task complexity.

chains, revealing substantial performance gaps: GPT-4 completes fewer than half of GTA queries once real tools and intermediate checks are enforced. MLGym casts the agent problem into a Gym environment for open-ended AI-research workflows, highlighting long-horizon planning and code execution and also without geospatial imagery Nathani et al. (2025).

**Remote Sensing Agents:** Recent efforts to extend LLM agents into EO have produced diverse tool-augmented pipelines, yet planning transparency, and step-level reasoning fidelity remain limited. Remote Sensing ChatGPT Guo et al. (2024) and RS-Agent Xu et al. (2024) represent early vision-language pipelines that chain pretrained detectors, segmenters, and geospatial utilities under GPT-based planners. However, they typically report only final answer accuracy, omitting structured ReAct-style trace evaluation or step-wise error attribution. TreeGPT and GeoMap-Agent Du et al. (2023); Huang et al. (2024) introduce domain-specific agents for forestry and geological mapping, respectively. While these systems operate over visual maps and structured visual inputs, they rely on template-grounded or qualitative responses and do not implement formal multi-step evaluation. UnivEARTH Kao et al. (2025), by contrast, employs a purely language-based framework that requires LLMs to generate valid Google Earth Engine (GEE) code, revealing that over 58% of completions fail to execute and that even the best agents answer only around 33% of geospatial queries correctly. Together, these works suggest that while EO agents can interface with rich toolsets, failures in tool selection, argument grounding, and spatial unit reasoning persist, underscoring the need for benchmarks that explicitly evaluate tool-level correctness alongside geospatial task outcomes.

**Evaluation Protocols:** Early benchmarks for tool-augmented LLMs, such as ToolBench Qin et al. (2023), APIBench Patil et al. (2023), and API-Bank Li et al. (2023), primarily evaluate single-step tool usage in synthetic or isolated API call settings. While useful for measuring basic tool and argument prediction, these setups lack support for multi-tool planning, intermediate tracking, or long-horizon reasoning. To address these limitations, GTA Wang et al. (2024) presents a tightly

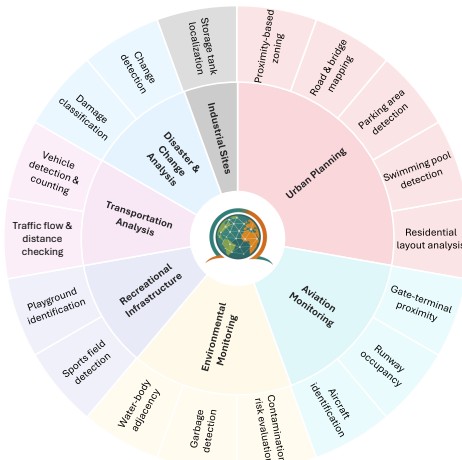
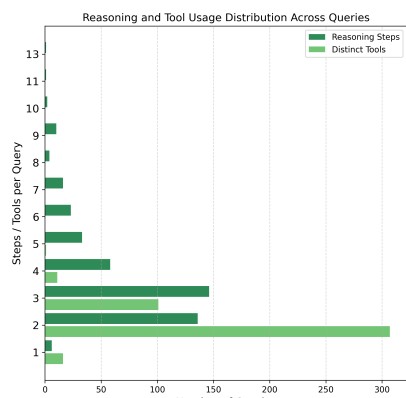

(a) **Use case taxonomy.** The benchmark spans seven major domains: Urban Planning, Disaster Assessment & Change Analysis, Environmental Monitoring, Transportation Analysis, Aviation Monitoring, Recreational Infrastructure, and Industrial Sites. Each domain includes representative task types requiring multimodal reasoning, spatial analysis, and tool-augmented execution.

(b) **Reasoning vs Tool Usage.** The barplot visualizes the complexity of agentic reasoning compared to the diversity of tool invocation across various tasks, highlighting different interaction and logical depth. The horizontal axis indicates the number of queries showing a given tool usage count. Most queries utilize 2-5 tools.

Figure 2: Overview of task domains and reasoning-tool interaction characteristics in ThinkGeo.

scoped yet richly instrumented benchmark requiring sequential tool usage across perception, logic, operation, and generation modules. GTA adopts a ReAct-style interface and introduces fine-grained supervision for each agent step, reporting metrics like ToolAcc, ArgAcc, StepAcc, and final answer correctness, thereby uncovering latent failure modes in tool selection and planning. Complementing this, MLGym Nathani et al. (2025) reframes agent evaluation as multi-task episodic learning within a Gym-style environment, simulating end-to-end ML workflows (e.g., training, evaluation, reporting) that demand persistent memory and adaptive behavior.

## 3 THINKGEO BENCHMARK

In this section, we present the design of **ThinkGeo**, a benchmark designed to evaluate tool-augmented LLM agents in the context of remote sensing. **ThinkGeo** focuses on spatially grounded reasoning tasks that require agents to interpret optical EO imagery, plan multi-step tool usage, and produce geospatially coherent outputs. We describe our core *design goals*, define the *use case categories* that span both primary and supporting remote sensing tasks, detail the *query construction pipeline*, and provide a summary of the *integrated datasets* and task coverage.

### 3.1 DESIGN GOALS

**Geospatial Reasoning:** Tasks are modeled after real-world use cases in environmental monitoring and disaster response. Queries reflect practical challenges such as measuring metre-scale distances, counting structures within spatial buffers, and identifying features of damaged buildings. These tasks require fine spatial fidelity, unit-based reasoning, and visual attribute grounding, capabilities often overlooked in existing benchmarks such as GAIA Mialon et al. (2023) or ToolBench Qin et al. (2023).

**Step-Implicit Tool Use:** Unlike benchmarks where tool use is predefined or explicitly mentioned (e.g., APIBench Patil et al. (2023)), ThinkGeo presents *step-implicit, tool-implicit* queries. Prompts do not reference tools by name; agents must infer which modules (e.g., perception, logic, operation) are needed and in what order. This design promotes true agentic planning and aligns with ReAct-style decision traces as used in GTA Wang et al. (2024).

Figure 3: End-to-end dataflow for constructing the ThinkGeo benchmark. We begin with expert-curated samples from remote sensing datasets, guided by scenario-specific query design templates. Human annotators inspect images and generate ReAct-style multi-step queries using a semi-automated GPT-powered interface. Each query is validated via expert review and script-based consistency checks. Invalid cases are manually refined. The final dataset consists of JSON-formatted ReAct traces grounded in satellite or aerial imagery.

## 3.2 USE CASE CATEGORIES

ThinkGeo is organized into seven primary categories, each reflecting critical application domains within the remote sensing ecosystem. These include *Urban Planning*, *Disaster Assessment & Change Analysis*, *Environmental Monitoring*, *Transportation Analysis*, *Aviation Monitoring*, *Recreational Infrastructure*, and *Industrial Sites*. Fig. 2a shows the use case taxanomy. Each category encapsulates a range of spatially grounded, tool-invoking subtasks inspired by operational workflows in urban analytics, environmental science, and infrastructure planning:

- *Urban Planning* tasks involve residential layout analysis, swimming pool and parking area detection, road and bridge mapping, accessibility assessment, and proximity-based zoning.

- *Disaster Assessment & Change Analysis* includes multi-temporal damage comparison across disaster events like floods, hurricanes, wildfires, and volcanoes, featuring change detection, categorical damage classification (e.g., no-damage, minor, major, destroyed), area-based summaries, and quadrant-level spatial reports.

- *Environmental Monitoring* spans water-body adjacency, garbage and construction waste detection, contamination risk evaluation, and agricultural land-use impact assessments.

- *Transportation Analysis* covers vehicle detection and counting, heading direction estimation, traffic flow characterization, and distance-based safety checks across roads and intersections.

- *Aviation Monitoring* includes aircraft identification and categorization, runway occupancy, gate-terminal proximity analysis, and airfield layout planning.

- *Recreational Infrastructure* tasks address playground identification (e.g., basketball, baseball, tennis, and football fields), orientation detection, and coverage estimation.

- *Industrial Sites* focus on storage tank localization, diameter and area measurement, and spatial relation mapping to adjacent operational zones.

These categories serve as testbeds for evaluating diverse capabilities such as multimodal reasoning, fine-grained spatial understanding, tool composition, and temporal change detection. By covering both canonical and underexplored use cases, ThinkGeo supports a systematic, application-driven evaluation of agentic LLM pipelines for real-world geospatial intelligence.

## 3.3 QUERY CONSTRUCTION PIPELINE

To evaluate the capabilities of agentic systems in solving realistic remote sensing problems, we curate a diverse set of *complex queries*, defined as prompts that are concise and natural for humans

but require agents to perform multi-step reasoning across multiple tools. These queries cannot be answered by the invocation of a single tool in isolation and instead test the agent's ability to plan and compose a coherent sequence of actions. We implement a semi-automated query generation pipeline.

**Step 1: Data Sampling & Guidelines.** We begin by curating high-quality samples from diverse RS datasets. Domain experts provide task-specific guidelines and generate initial reference queries to bootstrap the selection process. **Step 2: Authoring ReAct Format.** Using these guidelines, annotators manually inspect imagery, identify key objects and spatial relationships, and construct natural language queries following the ReAct format Yao et al. (2023). This involves composing a user query that implies multi-step reasoning, manually annotating missing elements, and generating a semi-structured dialog trace (thoughts, tool calls, observations, answers).

The process is supported by a script built on the OpenAI GPT API. This script leverages per-image metadata (e.g., object types, GSD, bounding boxes) and tool definitions to generate diverse, tool-requiring prompts. Used prompts in the query construction pipeline are given in the supplementary material. **Step 3: Validation.** All generated samples are verified through a two-stage validation protocol. First, expert reviewers assess the semantic correctness, relevance, and alignment with the toolset. Second, we apply script-based checks to validate tool argument consistency, dialog structure, and completeness. Invalid samples are manually refined and corrected before inclusion in the final dataset. Fig. 3 shows the end-to-end dataflow for constructing our ThinkGeo benchmark.

**Additional Details.** Beyond the core steps described above, the query construction pipeline incorporates several design elements to ensure scale, diversity, and inference robustness:

- **Query Diversity:** For each image, we generate 1 to 5 distinct queries that vary in spatial relationships, counting logic, or temporal comparisons, ensuring broad coverage of tool use compositions and reasoning patterns within the same scene context. As illustrated in Fig 2b, these queries span a range of agentic reasoning complexity and tool invocation diversity, capturing variations in interaction depth and logical structure across tasks.

- **Difficulty Annotation:** We categorized the queries into easy and hard levels based on two criteria: the number of complex keywords present and the number of reasoning steps required. Queries containing terms such as "estimate," "compare," "distribution," "count," "area," "how many," "orientation," and "proximity," along with more reasoning steps, were considered harder. To organize the queries, we sorted them based on the count of complex keywords and the number of steps. Queries appearing earlier in the sorted list, with fewer complex keywords and shorter reasoning steps, were labeled as easy, while the rest were classified as hard. This sorting strategy provides a simple yet effective way to separate queries by their semantic and procedural complexity.

- **Inference-Aligned Prompting:** Prompts are designed such that the agent must recover the reasoning chain without relying on field names or explicit tool indicators, promoting alignment with real-world, instruction-following behavior. This adheres to the tool-implicit design philosophy established in agentic benchmarks like GTA Wang et al. (2024).

This modular structure supports robust and scalable generation of diverse queries, enabling ThinkGeo to serve as a high-coverage benchmark for multimodal, tool-augmented RS agentic systems.

### 3.4 SOURCE RS DATASETS

To construct the ThinkGeo benchmark, we leverage a diverse set of publicly available remote sensing datasets (Table 2) spanning various domains: DOTA Xia et al. (2018), NWPU-VHR-10 Bian (2023), UCAS-AOD Zhu et al. (2015), and iSAID Waqas Zamir et al. (2019) support transportation and aviation-related tasks; FloodNet Rahnemoonfar et al. (2021) and xBD Gupta et al. (2019) contribute flood-specific and temporal disaster imagery; AID Xia et al. (2017) covers urban and industrial scenes; and the Global Dumpsite Dataset Sun & Yin (2023) addresses environmental monitoring. To expand transport and object-specific coverage, we additionally incorporate SSDD Zhang et al. (2021) for maritime ship monitoring, SADD Zhang et al. (2022) for aviation and aircraft monitoring, and SIVED Lin et al. (2023) for ground vehicle detection. Original images are reused, while task-specific annotations are added where the datasets lack required labels.

Table 2: Remote sensing datasets used as image sources in the construction of the ThinkGeo benchmark. These datasets span a wide range of applications, sensor resolutions, and annotation types. Notably, the agentic tasks defined on these images are newly annotated bottom-up.

| Name | Tasks | Annotation Type | Sensor (Res) | Year |
|---|---|---|---|---|
| **Optical RGB Datasets** | | | | |
| DOTA Xia et al. (2018) | Monitoring Transport, Aviation, Infrastructure | GSD, B-Box, Category | (0.1–1)m/px | 2021 |
| NWPU-VHR-10 Bian (2023) | Monitoring Transport, Aviation, Infrastructure | B-Box, Category | (0.5–2)m/px | 2023 |
| UCAS-AOD Zhu et al. (2015) | Monitoring Transport, Aviation | B-Box, Category | (0.5–2)m/px | 2015 |
| AID Xia et al. (2017) | Urban Planning, Monitoring Transport, Industr. Sites | B-Box, Category | (0.2–2)m/px | 2017 |
| iSAID Waqas Zamir et al. (2019) | Monitoring Transport | GSD, B-Box, Seg. Map, Pixel Count | (0.1–1)m/px | 2019 |
| xBD Gupta et al. (2019) | Disaster Assessment & Change Analysis | GSD, B-Box, Category, Pixel Count | (1–3.5)m/px | 2019 |
| FloodNet Rahnemoonfar et al. (2021) | Urban Planning, Disaster, Transport Analysis | GSD, B-Box, Category, Seg. Map, Pix. Count | (0.015–0.02)m/px | 2020 |
| Global-Dumpsite Sun & Yin (2023) | Environmental Monitoring | B-Box, Category | (0.3-0.8)m/px | 2023 |
| **SAR Datasets** | | | | |
| SSDD Zhang et al. (2021) | Monitoring Transport | B-Box, Category | (1–15)m/px | 2021 |
| SADD Zhang et al. (2022) | Aviation | B-Box, Category | (0.5–3)m/px | 2022 |
| SIVED Lin et al. (2023) | Monitoring Transport | B-Box, Category | (0.1-0.3)m/px | 2023 |

# 4 Tool Suite and Evaluation

**Task Format:** Each task is posed as a step and tool-implicit query, requiring the agent to reason and respond in a ReAct-style format Yao et al. (2023). Agents autonomously generate thought steps, select tools from a predefined set, format arguments, and produce final answers, evaluating spatial reasoning, planning, and multi-step execution grounded in remote sensing imagery.

**Tool Categories:** ThinkGeo extends the AgentLego framework AgentLego Contributors (2023) with two additional tools: `ChangeDetection` Irvin et al. (2024) for multi-temporal remote sensing analysis and `SegmentObjectPixels` Kirillov et al. (2023); Li et al. (2022) for segmentation and pixels counting. The toolset is organized into three functional categories: *Perception* (e.g., `TextToBbox`, `ChangeDetection`), *Logic*, and *Operation*, supporting object localization, spatial reasoning, and interactive annotation. Logic tools (e.g., `Calculator`, `Solver`) support numerical reasoning, distance calculations, and spatial comparisons. Operation tools (e.g., `DrawBox`, `GoogleSearch`) facilitate visual annotation and output formatting. This categorization supports fine-grained evaluation (e.g., tool-category performance) and structured analysis of planning behavior across spatial, logical, and domain-specific subtasks.

**Evaluation Methodology:** We adopt the evaluation framework of GTA Wang et al. (2024) for step-by-step metrics, including instruction-following (InstAcc), tool selection (ToolAcc), argument correctness (ArgAcc), and summary generation (SummAcc), to assess agent behavior. While GTA computes final answer accuracy (AnsAcc) using deterministic string matching, this can misclassify predictions due to variations in phrases. To mitigate this, we introduce LLM-as-a-judge: curated

Table 3: Evaluation results across models on the ThinkGeo benchmark. The table reports step-by-step (left) and end-to-end evaluation results (right), including tool-type accuracy (P: Perception, O: Operation, L: Logic), Ans. (final answer), and answer accuracy under image grounding (Ans_I). Overall, GPT4 family performs the best.

| Model | Step-by-Step Metrics | | | | End-to-End Metrics | | | | |
|---|---|---|---|---|---|---|---|---|---|
| | Inst. | Tool. | Arg. | Summ. | P. | O. | L. | Ans. | Ans_I |
| GPT-4o | *73.31* | *63.75* | *33.31* | 59.15 | **87.05** | *76.68* | **67.88** | **11.51** | **20.02** |
| GPT-4-1106 | **82.44** | **73.21** | **37.74** | *64.22* | 79.91 | 69.15 | 56.29 | 9.46 | *16.91* |
| Claude-3.7-Sonnet | 21.35 | 26.21 | 0.33 | **66.19** | *85.16* | **85.93** | *64.41* | 8.95 | 11.42 |
| Qwen1.5-7b-chat | 23.44 | 8.67 | 3.26 | 45.17 | 4.94 | 37.25 | 20.98 | 6.91 | 6.74 |
| Qwen2.5-7b-Instruct | 57.38 | 45.63 | 18.84 | *63.10* | 22.61 | 36.36 | 26.70 | 6.91 | 9.28 |
| InternLM3-8b-Instruct | 23.19 | 23.27 | 13.36 | 11.41 | 36.93 | 34.78 | 34.38 | 9.46 | 8.80 |
| LLaMA3-1-8b-Instruct | 44.58 | 35.98 | 13.49 | 54.66 | 54.87 | 34.20 | *56.41* | 7.16 | 8.12 |
| Phi-3-mini-4k-Instruct | 18.40 | 16.88 | 9.65 | 16.58 | 24.07 | 54.68 | 21.28 | 7.16 | 6.15 |
| Mistral-7B-Instruct-v0.2 | 18.51 | 16.56 | 0.00 | 44.57 | 31.38 | 14.74 | 40.51 | *11.25* | 5.15 |
| Yi-1.5-6B-Chat | 22.40 | 26.47 | 0.26 | 32.16 | 5.56 | 2.22 | 4.64 | 9.97 | 5.83 |
| Qwen3-8B (w/reasoning) | 20.98 | 13.36 | 3.26 | 50.95 | 59.45 | *70.37* | 33.53 | 7.67 | 8.68 |

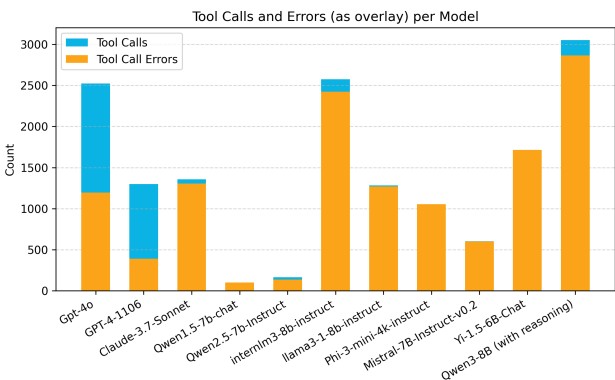

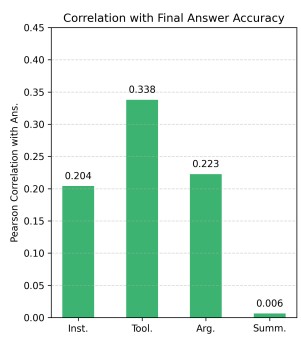

Figure 4: The plot illustrates the total number of tool calls made by each model and the corresponding number of tool call errors. The large discrepancy in open source models indicates a high rate of tool misuse. In contrast, models like GPT-4o demonstrate better tool invocation reliability.

Figure 5: This bar chart visualizes the Pearson correlation between step-by-step execution metrics and final answer accuracy on ThinkGeo.

evaluation questions per query and use 4o-mini to verify the correctness of the model's prediction. This offers a more reliable measure of task success, especially for multi-fact answers.

# 5 EXPERIMENTS & DISCUSSION

To assess the reasoning and tool-use capabilities of language models under real-world remote sensing scenarios, we conduct comprehensive evaluations on the ThinkGeo benchmark. Our benchmark poses multimodal and tool-implicit challenges that require agentic models to invoke tools across perception, operation, and logic categories. Unlike prior evaluations that rely on synthetic queries or shallow tool interactions, our benchmark emphasizes realistic queries grounded in satellite or aerial imagery and demands multi-step reasoning with spatial and numerical precision.

**Quantitative Analysis:** We evaluate a wide range of models, including GPT-4o Hurst et al. (2024), GPT-4-1106 Achiam et al. (2023), and several open-source variants (e.g., Qwen Hui et al. (2024); Bai et al. (2023), InternLM Cai et al. (2024), LLaMA3 Grattafiori et al. (2024), Phi Abdin et al. (2024), and Mistral Jiang et al. (2023)), in both step-by-step and end-to-end settings. The step-by-step mode evaluates intermediate stages such as instruction following (Inst.), tool selection (Tool.), argument formatting (Arg.), and summary generation (Summ.). The end-to-end mode measures performance on tool categories (P: Perception, O: Operation, L: Logic), final answer correctness (Ans.), and answer correctness under visual grounding (Ans_I). All evaluations and analyses presented in the paper are conducted on op-

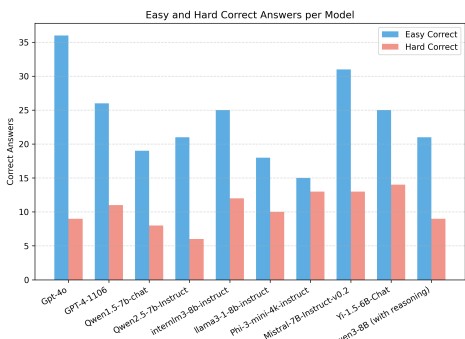

Figure 6: Number of correctly answered queries per model, categorized by difficulty level.

tical RGB imagery, ensuring consistency and comparability across models. To demonstrate the extension of our framework to additional modalities, we present SAR-based analysis in Appendix (section A.1). As reported in Table 3, GPT-4o and GPT-4-1106 achieve the strongest overall accuracy, reflecting superior planning and execution across multi-step tool chains. Most open-source models struggle with tool call formatting and argument prediction, resulting in significantly lower accuracy of the answers. Among all step-by-step metrics, tool selection has the highest correlation with final answer accuracy, underscoring its importance in agentic performance (Fig. 5).

**Tool Call & Error:** Tool calls and error rates highlight key gaps in agentic reliability. Proprietary models (GPT-4o, GPT-4-1106) show frequent tool use with relatively low error rates (47.3% and

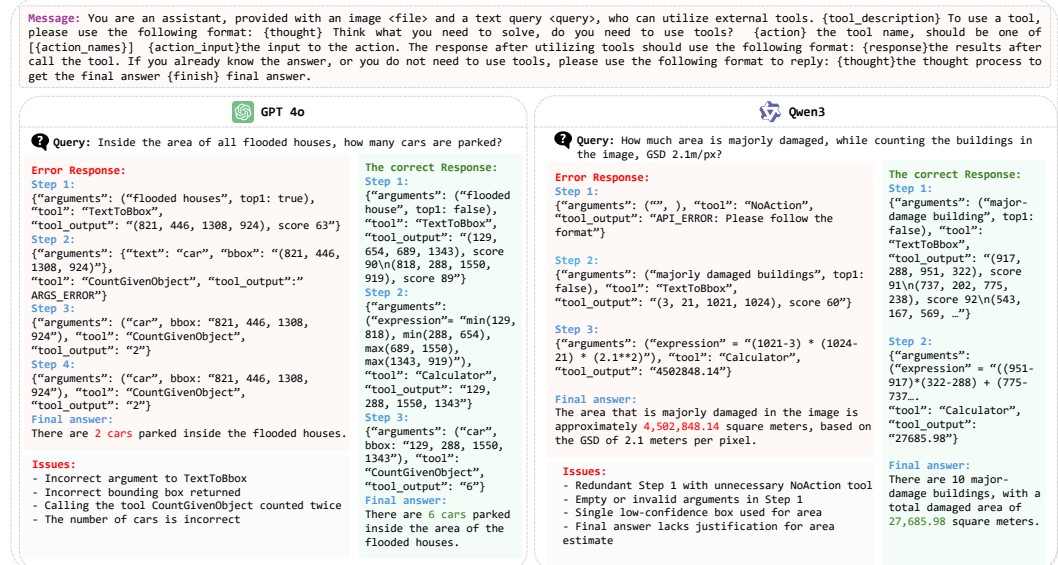

Figure 7: Examples of typical failure cases in GPT-4o and Qwen3 during ThinkGeo benchmark queries. On the left, GPT-4o struggles with incorrect argument formatting, misidentifies bounding boxes, redundantly invokes tools, and produces an incorrect final count. On the right, Qwen3 misuses tools (e.g., invoking `NoAction`), introduces redundant reasoning steps, and fails to provide spatial justification in its area estimate. In contrast, the correct responses illustrate structured reasoning with accurate spatial computation and coherent tool invocation.

30.0%), indicating strong tool-handling capabilities (Fig. 4). In contrast, open-source models (Qwen3-8B, InternLM3-8B, LLaMA3-8B) invoke tools aggressively but incur high error rates, reflecting poor execution control. Meanwhile, smaller models ( Qwen1.5-7B, Phi-3) often fail despite limited tool use, underscoring formatting and context alignment issues. These trends suggest that effective agent behavior hinges not just on tool access but on precise invocation and robust reasoning.

**Easy vs Hard Queries:** We analyze the performance of LLM agents on queries of varying difficulty levels, as defined in Section 3.3 Figure 6 presents a bar chart of correct responses per model, separated by difficulty level. The x-axis lists the evaluated models, while the y-axis indicates the count of correctly answered queries. Blue bars represent easy queries, and red bars denote hard ones. This analysis highlights a consistent performance gap across difficulty levels, emphasizing the increased challenges LLMs face when dealing with complex, multi-step reasoning tasks.

**Failure Analysis:** The qualitative examples in Fig. 7 illustrate common failure cases in multimodal agentic reasoning. GPT-4o, despite its high tool usage, struggles with incorrect argument formatting, misaligned bounding boxes, and redundant tool calls, resulting in inaccurate counts. Qwen3 frequently invokes unnecessary tools (e.g., NoAction), performs disconnected reasoning steps, and fails to justify numerical outputs with spatial context. These cases underline critical challenges in agent planning, such as argument misalignment, repeated tool misuse, and lack of unit-aware calculations, underscoring the need for precise reasoning across perception and logic modules in RS tasks.

## 6 CONCLUSION

We propose ThinkGeo, the first benchmark tailored specifically to evaluate tool-augmented LLM agents on real-world RS tasks. Since ThinkGeo grounds evaluation in high-resolution EO imagery, structured tool-use pipelines, and fine-grained reasoning annotations, it reveals critical gaps in current agent capabilities. In particular, our analysis shows room for improvement in spatial planning, temporal consistency, and domain-specific tool integration. Our extensive study across 486 tasks and multiple SoTA LLMs demonstrates that, while tools like segmentation and change-detection improve raw perception, true geospatial reasoning remains an open challenge. ThinkGeo aims to attract further efforts towards the development of next-generation multimodal agents that can seamlessly blend perception, planning, and execution in complex, spatially grounded RS and EO environments.

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

# A APPENDIX

This supplementary material provides extended analysis and additional results to support the main paper. It includes evaluations on the SAR data(A.1), and detailed error breakdowns (A.2). We also present representative samples across use case categories (A.3), tool usage distribution (A.4), and category-wise sample counts (A.5). Moreover, we report the human effort (A.6) involved in curating, generating, and verifying the proposed ThinkGeo benchmark. Additionally, we analyze the runtime performance of the evaluated models (A.7)

## A.1 SAR DATA EVALUATION

A set of 50 queries over 244 tool-reasoning steps utilize SAR imagery. This data is developed through the same rigorous, three-phase manual curation pipeline described above, ensuring identical depth and consistency in annotation; evaluation results are reported in Table 4. Proprietary models like GPT-4 variants lead in instruction and tool usage accuracy on SAR imagery, showing strong generalization in structured reasoning. However, final answer accuracy remains low ($< 10\%$) across all models, revealing that even the best proprietary models struggle with visual grounding in non-optical scenarios. This highlights a key gap: tool competence alone is not sufficient for reliable SAR understanding.

Table 4: Evaluation results across models on the ThinkGeo SAR benchmark. The table reports step-by-step execution metrics (left) and end-to-end evaluation results (right), including tool-type accuracy (P: Perception, O: Operation, L: Logic), Ans. (final answer), and answer accuracy under image grounding (Ans_I).

| Model | Step-by-Step Metrics | | | | End-to-End Metrics | | | | |
|---|---|---|---|---|---|---|---|---|---|
| | Inst. | Tool. | Arg. | Summ. | P. | O. | L. | Ans. | Ans_I |
| GPT-4o | 75.00 | 69.67 | **49.18** | 50.00 | **87.43** | **81.82** | 48.15 | 5.56 | 22.29 |
| GPT-4-1106 | **75.71** | **71.31** | 48.77 | 58.33 | 56.93 | 70.45 | 17.45 | 5.56 | 17.10 |
| Claude-3.7-Sonnet | 14.64 | 21.72 | 0.41 | **77.78** | 85.93 | 55.56 | **70.06** | **8.33** | **24.00** |
| Qwen1.5-7b-chat | 15.36 | 4.10 | 2.05 | 36.11 | 0.00 | 47.06 | 1.44 | 2.78 | 10.00 |
| Qwen2.5-7b-Instruct | 56.43 | 44.26 | 29.10 | 61.11 | 16.49 | 26.67 | 27.97 | 5.56 | 8.00 |
| InternLM3-8b-Instruct | 51.79 | 47.13 | 29.92 | 36.11 | 30.87 | 30.51 | 29.81 | 0.00 | 7.74 |
| LLaMA3-1-8b-Instruct | 40.71 | 34.43 | 20.08 | 55.56 | 56.38 | 25.35 | 46.06 | 2.78 | 8.00 |
| Phi-3-mini-4k-Instruct | 33.21 | 29.51 | 16.80 | 33.33 | 21.05 | 25.00 | 24.52 | 5.56 | 8.00 |

## A.2 ERROR ANALYSIS

The error analysis summarized in Tab. 5 provides insights into the types of prediction failures exhibited by different models on the ThinkGeo benchmark, categorized into planning and format-related errors. Planning errors, specifically NoAction, is prominent for models like GPT-4-1106 (84.87%), GPT-4o (95.82%), and Qwen1.5-7b (94.90%), indicating extra reasoning or summaries without delivering a final actionable response, i.e., either a tool call or an explicit answer. In contrast, models such as LLama3-1-8b and Qwen2.5-7b show high rates of "Invalid JSON" errors (82.89% and 84.44%, respectively), revealing difficulty in producing syntactically correct tool input formats even when the correct tool is selected. Among format-related errors, "Final Answer (SingleStep)" exhibits high rates: Phi-3-mini-4k (73.74%), Qwen1.5-7b (80.05%), and Qwen2.5-7b (74.94%), where models bypass intermediate reasoning and prematurely generate a final answer. In particular, GPT-4o and GPT-4-1106 show remarkably low error rates in the format category, demonstrating strong capabilities in structured reasoning. Overall, the results highlight that models face different challenges, some struggle with planning and step-by-step reasoning, while others have issues formatting tool inputs or following the required response structure.

## A.3 MORE SAMPLES BY USE CASE

Fig. 8 shows examples in which ThinkGeo queries prompt-driven agents to compose multiple tools from the available set. Each ReAct-style execution chain demonstrates spatial reasoning and multi-step decision making grounded in satellite imagery, reflecting the benchmark's focus on real,

Table 5: Breakdown of errors made by models on the ThinkGeo benchmark. Errors are grouped into planning and format-related categories (percentages are reported).

| Model | Planning | Format Errors | | | |
|---|---|---|---|---|---|
| | NoAction | Inv. JSON | Arg. Values | Tool Name | Final Ans. (SingleStep) |
| GPT-4o | 95.82 | 1.92 | 2.18 | 0.08 | 0 |
| GPT-4-1106 | 84.87 | 11.54 | 3.08 | 0.26 | 0 |
| Claude-3.7-Sonnet | 58.04 | 41.73 | 0.00 | 0.23 | 18.44 |
| Qwen1.5-7b | 3.06 | 94.90 | 0.00 | 2.04 | 80.05 |
| Qwen2.5-7b | 11.11 | 84.44 | 4.44 | 0.00 | 74.94 |
| InternLM3-8b | 57.37 | 33.92 | 0.62 | 7.80 | 44.44 |
| LLaMA3-1-8b | 13.01 | 82.89 | 0.16 | 3.94 | 14.08 |
| Phi-3-mini-4k | 56.36 | 40.61 | 0.09 | 2.94 | 73.74 |
| Mistral-7B | 49.17 | 40.67 | 0.17 | 10.00 | 53.13 |
| Yi-1.5-6B | 81.06 | 4.25 | 0.00 | 14.69 | 59.71 |
| Qwen3-8B | 81.34 | 18.52 | 0.07 | 0.00 | 4.32 |

tool-oriented geospatial problem solving. Additionally, we illustrate complete reasoning trajectories and grounding in queries in Fig. 10 and 11

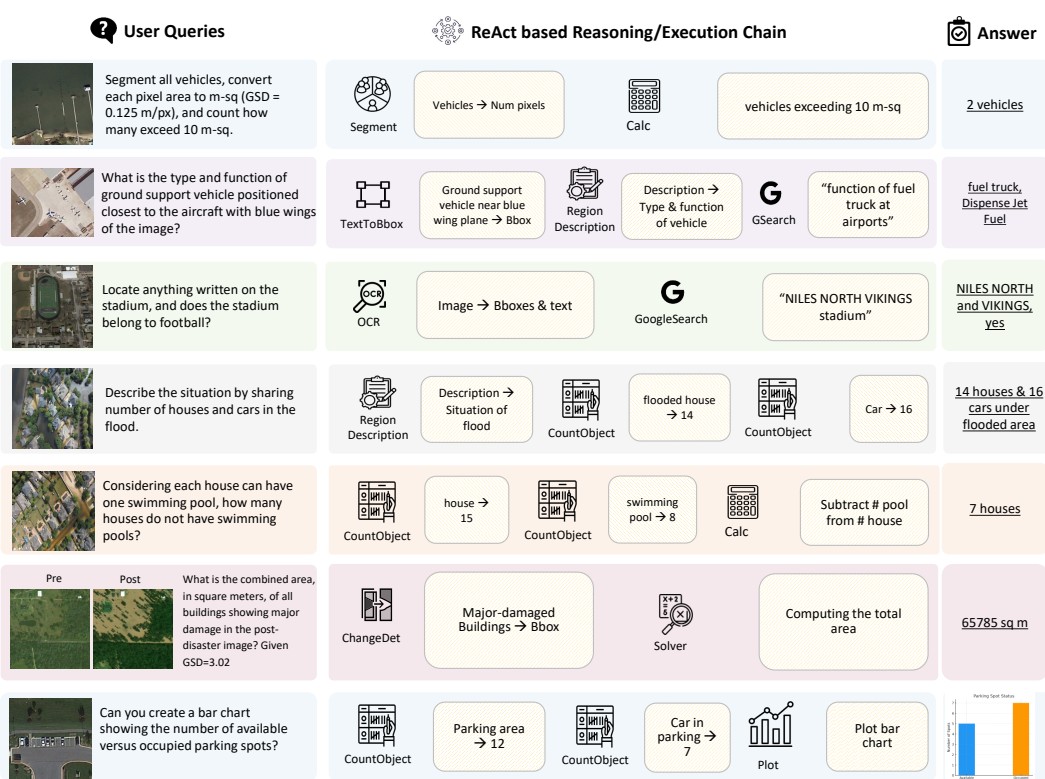

Figure 8: Representative examples from the ThinkGeo benchmark. Each row shows a user query (left), the corresponding ReAct-style execution chain involving tool calls (center), and the final answer (right).

## A.4 TOOL USAGE DISTRIBUTION

The distribution of tool usage across the ThinkGeo benchmark is shown in Fig. 9. The most frequently invoked tools are `Calculator`, `TextToBbox`, and `RegionAttributeDescription`, re-

Table 6: Category-wise count of easy and hard tasks, including totals.

| Main Category | Combined | Easy-level | Hard-level |
|---|---|---|---|
| Disaster Assessment & Change Analysis | 148 | 95 | 53 |
| Urban Planning | 71 | 48 | 23 |
| Transportation Analysis | 90 | 54 | 36 |
| Aviation Monitoring | 40 | 24 | 16 |
| Industrial Sites | 15 | 12 | 3 |
| Recreational Infrastructure | 41 | 28 | 13 |
| Environmental Monitoring | 31 | 20 | 11 |
| **Total** | **436** | **281** | **155** |

flecting the benchmark's emphasis on spatial computation, object localization, and attribute reasoning. Mid-frequency tools such as `CountGivenObject` and `ChangeDetection` support core analysis tasks, while tools such as `AddText`, `ObjectDetection`, and `OCR` are rarely used, indicating their narrower application scope. This spread highlights the diversity in tool reliance and the complexity of multi-step reasoning across geospatial scenarios.

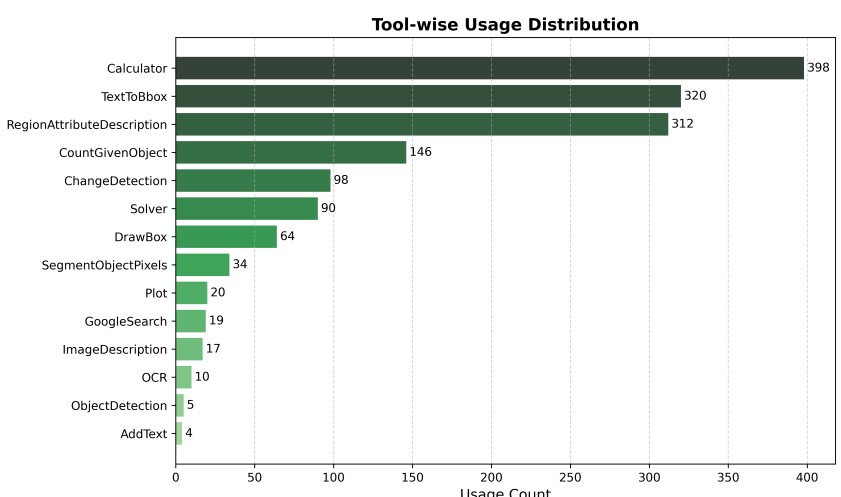

Figure 9: Tool-wise usage distribution in the ThinkGeo benchmark.

## A.5 CATEGORY-WISE SAMPLE STATISTICS

Tab. 6 summarizes the ThinkGeo benchmark by task category and difficulty level. Among the seven core domains, Disaster Assessment & Change Analysis contributes the largest share (148 queries), reflecting the complexity of temporal reasoning and damage interpretation. Urban Planning and Transportation Analysis are followed by diverse spatial reasoning tasks.

## A.6 HUMAN ANNOTATION EFFORT

The construction of the ThinkGeo benchmark involved a coordinated annotation effort totaling $\approx 500$ human hours, distributed across a 4-member team. This time was spent across three core stages:

**Sample Selection and Query Drafting (100 hours):** Annotators began by selecting high-quality image samples from diverse remote sensing datasets. Scenario-specific query design guidelines were developed per use case, and representative prompts were drafted for further guidance.

**Analyzing Image and ReAct Construction (280 hours):** Human annotators analyzed each image to understand the spatial and semantic context, composed task-relevant queries, and manually filled in missing annotations (e.g., object bounding boxes, counts) where necessary. Using a semi-automated

interface powered by `o4-mini`, ReAct-style interaction traces were constructed that included thought steps, tool invocations, and final answers, all grounded in the content.

**Validation and Finalization (120 hours):** Each ReAct trace was rigorously verified through manual review and script-based checks. This involved ensuring that all reasoning steps (thoughts), tool arguments, and outputs were present and logically coherent, and that the final answer could be derived correctly.

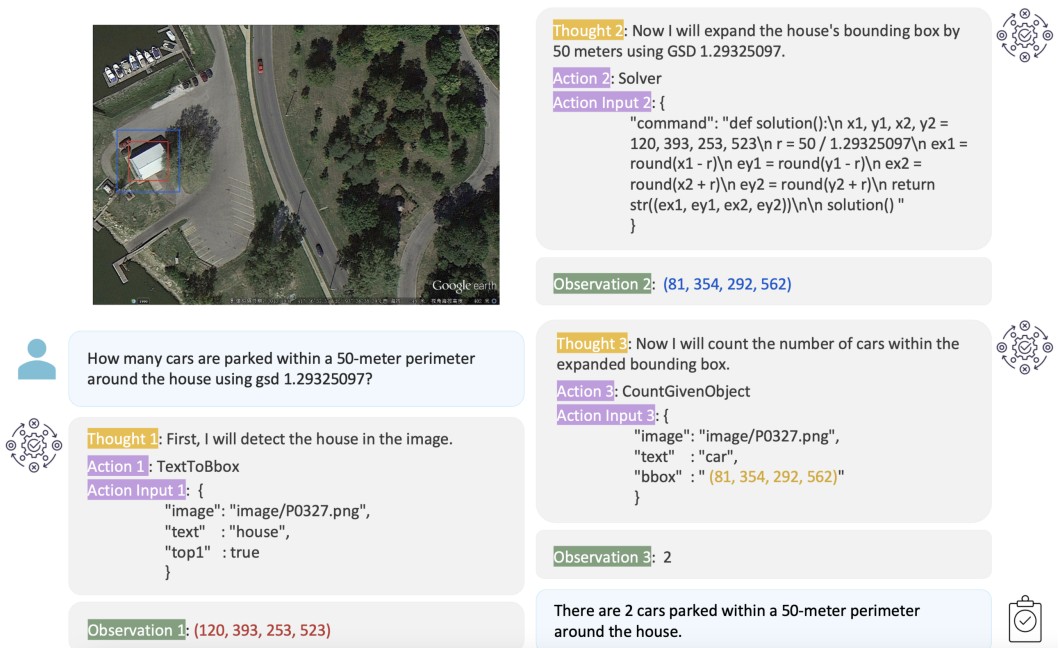

Figure 10: Example of a tool-augmented reasoning in ThinkGeo benchmark query. The illustration shows a multi-step task where the agent is required to combine perception, logic, and spatial reasoning tools in sequence. The figure highlights how queries are grounded in real remote sensing imagery and require tool-augmented reasoning chains to arrive at correct geospatial conclusions.

## A.7 RUNTIME ANALYSIS

To complement the evaluation results, we analyze the runtime performance of the evaluated models. Table 7 reports both the step-by-step latency (measuring average runtime per query) and the end-to-end pipeline latency (including LLM inference, tool calls, and result aggregation). These results highlight the computational cost of agentic reasoning for geospatial tasks.

Table 7: Average runtime per query across evaluated models. Step-by-step latency corresponds to the time required to execute all reasoning steps; end-to-end latency accounts for the entire pipeline execution.

| Model | Step-by-step Avg. (s/query) | End-to-end Avg. (s/query) |
|---|---|---|
| GPT-4o | 12.10 | 12.01 |
| GPT-4-1106 | 22.43 | 16.65 |
| Qwen1.5-7b-chat | 22.54 | 6.79 |
| Qwen2.5-7B-Instruct | 16.79 | 7.17 |
| InternLM3-8b-Instruct | 22.68 | 27.01 |
| LLaMA3-1-8b-Instruct | 13.16 | 14.78 |

While proprietary models such as GPT-4o and GPT-4-1106 offer relatively stable runtime profiles, open-source models often display higher variability across step-by-step and end-to-end latencies.

Notably, InternLM3-8b exhibits significant end-to-end overhead. Step-by-step evaluation is longer for some models, like Qwen1.5, despite appearing more incremental in nature. These models process each action prediction by re-encoding the entire history of previous steps as context. This cumulative context grows with each step, significantly increasing the load and inference time. By contrast, in the end-to-end setting, Qwen1.5 often generates answers with fewer tool calls or shorter reasoning traces, resulting in less overall computation.

These findings align with prior observations that current agentic pipelines continue to incur substantial latency Mialon et al. (2023). Reducing end-to-end response time remains an important open research direction for enabling real-time geospatial applications.

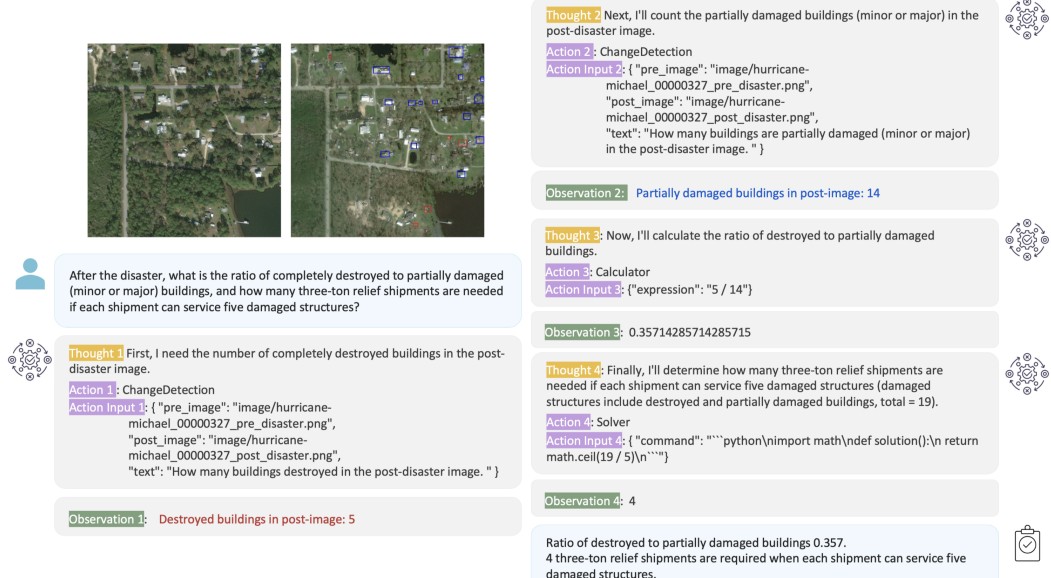

Figure 11: Example of change detection in ThinkGeo Post-disaster damage assessment and resource planning. The figure illustrates a task that requires the identification and comparison of temporal differences between pre-disaster and post-disaster imagery. The ReAct-style execution chain illustrates how agents must invoke perception and computation tools to quantify changes, such as structural damage.

**LLM Usage Statement:** We used large language models exclusively for polishing the writing and providing assistance during dataset curation. They were not involved in research ideation, model design, and analysis.

