# OpenReview forum: "ThinkGeo: Evaluating Tool-Augmented Agents for Remote Sensing Tasks"
_ICLR.cc/2026/Conference — ICLR 2026 Conference Withdrawn Submission_

### Official Review · Reviewer_xsCN · 2025-10-31

**Soundness:** 3
**Presentation:** 2
**Contribution:** 2
**Rating:** 2
**Confidence:** 4

**Summary:**

This paper presents ThinkGeo, a benchmark designed to evaluate *tool-augmented LLM agents* in remote sensing (RS) tasks. Unlike general-purpose benchmarks such as ToolBench, GAIA, or GTA, ThinkGeo emphasizes *spatial reasoning* grounded in satellite and aerial imagery, assessing how agents plan, invoke tools, and integrate perception, logic, and operation modules.
The benchmark covers 486 agentic tasks with 1,773 expert-verified reasoning steps across seven RS domains (urban planning, disaster assessment, environmental monitoring, transportation, aviation, recreation, and industry). It includes an executable tool suite (14 tools) and two evaluation modes (step-by-step and end-to-end). Experiments over diverse models (GPT-4o, Claude-3.7, Qwen2.5, LLaMA3, etc.) reveal large performance gaps in spatial grounding, argument formatting, and multi-step consistency.

**Strengths:**

1. Novel application domain: ThinkGeo meaningfully extends agentic evaluation into remote sensing, an underexplored yet practically critical field.
2. Comprehensive dataset design: 486 tasks annotated with fine-grained ReAct traces and human validation offer substantial scale and detail.
3. Structured tool taxonomy: The integration of perception, logic, and operation tools (e.g., `ChangeDetection`, `SegmentObjectPixels`) provides a modular evaluation of spatial and reasoning ability.
4. Clear methodology: The pipeline for query generation, validation, and dataset curation is carefully documented and reproducible.
5. Empirical breadth: Covers a wide range of LLMs and provides both quantitative metrics (InstAcc, ToolAcc, ArgAcc) and qualitative analyses (failure examples, tool call errors).

**Weaknesses:**

1. Despite the impressive engineering effort, the paper mainly adapts existing ReAct-style benchmarks (e.g., GTA, ToolBench) to a new domain. It does not introduce fundamentally new evaluation principles or algorithms—its novelty lies primarily in the application context.
2. The benchmark remains evaluation-oriented rather than analysis-oriented. While the authors present tool accuracy and reasoning consistency, there is minimal exploration of why models fail (e.g., visual grounding vs. reasoning gaps) or how agentic design can mitigate these issues.
3. The benchmark’s query generation pipeline depends on human experts and semi-automated GPT scripting. This makes it difficult to replicate or extend without significant effort, and introduces potential annotator bias. A more data-driven or semi-synthetic generation strategy would enhance scalability.
4. The benchmark evaluates models only on remote-sensing tasks but does not verify whether improvements transfer to non-RS domains. Without such comparison, it is unclear whether ThinkGeo assesses *general agentic reasoning* or simply *domain familiarity*.
5. The design choices—such as the selected 14 tools, ReAct format, and evaluation metrics—are taken as fixed. There is no ablation on how task difficulty, tool diversity, or image modality (RGB vs. SAR) affect performance.
6. The paper adopts “LLM-as-a-judge” evaluation for correctness but provides little evidence of its reliability. No inter-rater agreement or validation against human experts is reported. This weakens the credibility of reported accuracy metrics.
7. Much of the main text reads like a technical report rather than an academic analysis. Figures and tables convey results clearly, but interpretive discussion (e.g., why GPT-4o outperforms open-source models, or what features drive performance) is limited.

**Questions:**

1. Please quantify the reliability of your LLM-as-a-judge system. How consistent are its decisions with human evaluation? Reporting an inter-annotator agreement or accuracy over a labeled subset would strengthen the methodological validity.
2. Conduct an ablation varying (a) the number of tools available and (b) the task difficulty (easy vs. hard). Does model performance degrade linearly with reasoning steps or tool diversity?
3. Evaluate a small subset of models (e.g., GPT-4o, Qwen2.5) on cross-domain tool reasoning (such as GAIA or GTA tasks) to assess whether ThinkGeo captures domain-specific or general agentic competence.
4. The qualitative failures in Fig. 7 are informative; please complement them with a quantitative breakdown by error type (argument errors, redundant steps, wrong bounding boxes, etc.) across all models.
5. Appendix A.1 briefly reports SAR results but lacks discussion. Please analyze why optical-trained agents fail on SAR—e.g., due to missing spectral priors or inappropriate tool arguments.

---

### Official Review · Reviewer_KRQH · 2025-10-31

**Soundness:** 3
**Presentation:** 3
**Contribution:** 3
**Rating:** 4
**Confidence:** 4

**Summary:**

This paper presents a benchmark for geospatial agents on remote sensing tasks. The authors collected 486 tasks  spanning a set of different applications from urban planning to change analysis and aviation monitoring etc, and grounded the answers using satellite or arial imagery. To succeed an agent has to crrectly reason through the question and the images to find specific answers. The dataset is well constructed including with expert reasoning steps. The authors built an agent to evaluate the benchmark on a set of different models with reasonable accuracy on frontier models on the step by step evaluations but weak overall accuracy on the tasks.

**Strengths:**

The strength of the paper is the benchmark itself, which appears carefully put together, as well as teh sets of tool calls need to answer the questions. The expert review on each question evaluates the accuracy against the entire task suite.

**Weaknesses:**

The difficulty of this benchmakr is hard to understand. It was constructed from a standard set of publically available remote sensing datasets, with questions posed by "experts" about the datasets being the core of the benchmark.  If a model or a person could successfully solve all of the problems in the benchmark, what level of expertise do they have?

The expert evaluation is welcome but this makes it difficult for this to serve as a reproducible benchmark -- we aren't given much information about the experts and their level of expertise, but from the tasks given the questions in the benchmark seem straightforward to answer even for a nonexpert.

**Questions:**

What is the difficulty of the benchmark, measured in terms of skills of expert human. Easy and hard are relative terms and are not well calibrated. If a system could perform well on this benchmark what capability would it have?
To what extent do the tools you are providing to the agent affect the quality of the answer

---

### Official Review · Reviewer_aLv1 · 2025-11-01

**Soundness:** 4
**Presentation:** 4
**Contribution:** 3
**Rating:** 6
**Confidence:** 3

**Summary:**

This paper introduces ThinkGeo, a new benchmark designed to evaluate tool-augmented LLM agents on remote sensing tasks. Unlike existing benchmarks such as GAIA, ToolBench, or GTA that focus on general-purpose reasoning or synthetic multimodal setups, ThinkGeo specifically targets spatially grounded, real-world Earth observation applications. The benchmark includes: 1. 486 agentic tasks with 1,773 expert-verified reasoning steps, 2. a suite of 14 tools for perception, logic, and spatial operations (e.g., object detection, segmentation, calculator), 3. step-by-step and end-to-end evaluation metrics (e.g., ToolAcc, ArgAcc, AnsAcc), 4. comparative evaluation of leading models (GPT-4o, Claude-3.7, Qwen2.5, LLaMA-3, etc.). The authors show that even top models struggle with tool argument formatting, spatial grounding, and multi-step planning, highlighting open challenges in agentic reasoning for geospatial tasks.

**Strengths:**

The authors introduce a novel agentic benchmark for the remote sensing domain. While the benchmark is very domain specific, it requires skills such as fine-grained visual reasoning, geospatial reasoning, and tool selection, which are generalizable to a wide variety of domains. The benchmark curation pipeline is rigorous with expert validation and error analyses. Inclusion of finegrained step-wise evaluation is also not common of agentic baselines.

**Weaknesses:**

The paper could benefit by including more details on benchmark curation and evaluation. See questions.

**Questions:**

The authors use gpt-4o for LLM-as-a-judge. Have you calibrated the juge across different models to verify concordance with human judgment / prevent bias toward GPT-family outputs?

In Section 4 evaluation methodology, including concise descriptions of InstAcc, ToolAcc, etc would make the paper more self-contained.

The benchmark relies on publicly available datasets. Is there risk of data leakage if pretrained models have seen these images before?
The datasets also heavily feature urban and transportation-heavy scenes. Does this create bias against rural and vegetation-dominant regions?

"Queries appearing earlier in the sorted list, with fewer complex keywords and shorter reasoning steps, were labeled as easy, while the rest were classified as hard." Could the authors clarify what "fewer" and "shorter" mean here? Was an explicit threshold used?

Table 3 reports very close numbers for some metrics. Is it possible to report a measure of spread (SD, confidence interval) etc to help contextualize these difference?

Could the authors include average per query costs for some common closed source models?

---

### Note · Authors · 2025-11-13

I have read and agree with the venue's withdrawal policy on behalf of myself and my co-authors.